# The human posterior parietal cortices orthogonalize the representation of different streams of information concurrently coded in visual working memory

**Yaoda Xu** [ORCID] *

Department of Psychology, Yale University, New Haven, Connecticut, United States of America

* yaoda.xu@yale.edu

**Citation:** Xu Y (2024) The human posterior parietal cortices orthogonalize the representation of different streams of information concurrently coded in visual working memory. PLoS Biol 22(11): e3002915. https://doi.org/10.1371/journal.pbio.3002915

**Data Availability Statement:** All data files and analysis scripts contributed to this study are available at https://osf.io/8rbkh/.

## Abstract

The key to adaptive visual processing lies in the ability to maintain goal-directed visual representation in the face of distraction. In visual working memory (VWM), distraction may come from the coding of distractors or other concurrently retained targets. This fMRI study reveals a common representational geometry that our brain uses to combat both types of distractions in VWM. Specifically, using fMRI pattern decoding, the human posterior parietal cortex is shown to orthogonalize the representations of different streams of information concurrently coded in VWM, whether they are targets and distractors, or different targets concurrently held in VWM. The latter is also seen in the human occipitotemporal cortex. Such a representational geometry provides an elegant and simple solution to enable independent information readout, effectively combating distraction from the different streams of information, while accommodating their concurrent representations. This representational scheme differs from mechanisms that actively suppress or block the encoding of distractors to reduce interference. It is likely a general neural representational principle that supports our ability to represent information beyond VWM in other situations where multiple streams of visual information are tracked and processed simultaneously.

## Introduction

Extensive research has documented that not only can the content of visual working memory (VWM) be decoded from human fMRI voxel response patterns during the delay period across multiple brain regions, including early visual areas (EVCs) and posterior parietal cortex (PPC) (for reviews, see [1–4]), but critically, the content of VWM can be recovered under distraction, with PPC showing greater distraction-resilience than EVC [5–7]. Notably, distractor information can also be decoded across EVC and PPC, either with near-ceiling performance (for face and gazebo distractors; [5]) or as good as the targets (for grating distractors; [6]), indicating concurrent representations of both target and distractor information in VWM. Thus, active suppression/filtering mechanisms that have traditionally been observed to combat distraction,

**Funding:** Research reported in this publication was supported by the National Eye Institute of the National Institutes of Health (NIH, https://grants.nih.gov/) under Award Number R01EY030854 to YX. The content is solely the responsibility of the authors and does not necessarily represent the official views of NIH. NIH had no role in study design, data collection and analysis, decision to publish, or preparation of the manuscript.

**Competing interests:** The authors have declared that no competing interests exist.

**Abbreviations:** BOLD, blood-oxygen-level-dependent; EVC, early visual area; FIR, finite impulse response; LOT, lateral occipitotemporal; MDS, multidimensional scaling; OTC, occipitotemporal cortex; PPC, posterior parietal cortex; RDM, representational dissimilarity matrix; ROI, region of interest; SMS, simultaneous multislice; VOT, ventral occipitotemporal; VWM, visual working memory.

such as those involving distractor suppression or blocking [8,9], appear to be either less effective or may not have been evoked. This raises an intriguing question: If distractors are not effectively suppressed or blocked in VWM, how then are we able to resist distraction?

To match our daily visual experience involving a constant and ever-changing influx of visual input, attention-grabbing distractors were shown continuously during the VWM delay period in prior studies [5,6]. This is unlike prior neurophysiological studies in which a distractor was briefly shown during delay to temporarily disturb target representation which was then restored [10–12]. In a continuous distractor presentation paradigm, the saliency of the distractors makes them difficult to suppress or block. The key to distractor resistance in this case thus may not be in suppressing or blocking them, but instead in our ability to untangle and separate VWM target and distractor representations. One way to do so is to orthogonalize these representations. Here in the state space formed by neuronal population responses [13], even though the joint target and distractor representation can vary with the presence/absence of distraction and the identity of the distractors shown, with orthogonal target and distractor representations target information can still be read out independently of distraction by downstream regions, effectively combating distraction. An orthogonal representation thus untangles and separates target and distractor representations, forming distractor-tolerant target representations, while simultaneously accommodating the representation of distractors (here, the term "entanglement" is borrowed from the object processing literature as described below (e.g., [14,15]), in which untangled representations would enable independent readout of the different representations by downstream regions, whereas tangled representations would not; the usage of the term "entanglement" here thus differs from its usage in other contexts, such as [16]).

Using an orthogonal representational scheme to untangle different streams of information has previously been documented to play a critical role in the extraction of object identity information in the primate occipitotemporal cortex (OTC) during object recognition. Specifically, given that the coding of nonidentity object features, such as position, size, and viewpoint, cannot be blocked as they are an integral part of the visual input for an object, OTC visual processing develops the ability to untangle object identity and nonidentity features in its neural representational space [17–24]. This allows independent access to object identity and enables object recognition despite changes in viewing conditions. Such a representational scheme has been hailed as the hallmark of ventral visual processing [14,15] and is notably absent in convolutional neural networks trained for object classification [22].

Orthogonalizing information for independent access can thus serve as an effective representational mechanism to reduce interference when multiple streams of information are coded together. A key feature of this geometry is that a linear neural response decoder trained to classify a pair of objects in one condition can generalize its performance to another condition. Indeed, previous studies have verified this and used it to document the untangling of object identity and nonidentity features in OTC [17–19]. The same approach is thus used here to test if target and distractor orthogonalization exists in the human brain as an effective mechanism to combat distraction in VWM. Besides using cross-decoding, to provide a direct measure of the representational geometry, the present study also directly visualizes the target and distractor representational space and measures the representational angles to bring evidence convergent with that obtained from cross-decoding.

Prior research has described the sensory nature of VWM representation in human EVC [1–3]. This could potentially put VWM targets and distractors in the same representational subspace, making it harder to untangle them. PPC, on the other hand, separates visual representations by task goals [25–28]. With VWM targets and distractors assigned different task goals (i.e., task-relevant versus task-irrelevant), PPC may code them in different

representational subspaces, potentially enabling easy untangling of targets and distractors. Indeed, PPC has been shown to exhibit greater distractor resistance than EVC [5–7]. Existing literature thus predicts that, should target and distractor orthogonalization exists in VWM, it would be more likely to be found in PPC than in EVC or OTC.

Besides distractors, another source of interference to a target stored in VWM is other concurrently stored targets. When targets are shown at different locations, the prevalence of location coding throughout the primate brain [29] allows targets to be coded by distinctive neuronal populations selective for different locations, resulting in orthogonal target representation during the delay period [30]. However, it is unclear how targets may be differentiated when location coding is unavailable, such as when targets are shown sequentially at the same location and when the presentation order is task-irrelevant. If information orthogonalization is a general and effective neural mechanism deployed to combat interference in VWM, then it may be utilized to differentiate multiple targets retained in VWM even when they share the same location. Here, the joint target-target representation may differ depending on the exact set of targets retained; however, as long as target representations are orthogonalized, information about a given target can still be read out independently of the other retained targets by downstream regions, effectively combating interference among the concurrently held targets in VWM. Using the same cross-decoding approach and direct angle measurement of the representational space, the second experiment of this study tested the existence of representation orthogonalization for 2 concurrently stored targets in VWM sharing the same location. While PPC may exhibit such coding given its greater ability to resist distraction than OTC, with feedback from association areas such as PPC driving VWM representation in OTC ([31]; see also [12,32–34]), similar representation could also exist in both PPC and OTC.

Together, the 2 experiments of the present study aim to examine whether or not target representations in VWM are modulated by the distractors shown during the delay and by other concurrently held targets in VWM. An effect of modulation would indicate the existence of interactive and entangled representations, whereas an absence of such an effect would show the existence of independent and untangled representations. To accomplish this, fMRI pattern decoding was used to document if a classifier trained in one condition may generalize its performance to another condition; decoding results were also used to directly visualize the VWM object representational space.

## Results

### Untangling target and distractor representations in VWM

To test if targets and distractors form orthogonal and independent representations in VWM as an effective mechanism to combat distraction, in Experiment 1, a linear decoder was trained to classify the fMRI response patterns of a pair of target objects in one delay condition; the decoder's performance was then tested on the same object pair both in the same and in a different delay condition (Fig 1). If representation orthogonalization exists in VWM, then a complete generalization of the decoder's performance is expected across different distractor conditions. However, if targets and distractors are entangled and interact in VWM, then an incomplete generalization is expected.

To do so, 14 human participants (9 females) retained real-world objects in VWM, either with or without distraction during the delay (randomly intermixed), while their brain responses were recorded with a 3T MRI scanner across 2 scan sessions (Fig 1A). The same 4 types of objects (i.e., bike, couch, hanger, and shoe) served as either targets or distractors in different trials to simulate real-world vision where the same object may be either task-relevant or irrelevant depending on behavioral goals (Fig 1B). To ensure the retention of visual code in

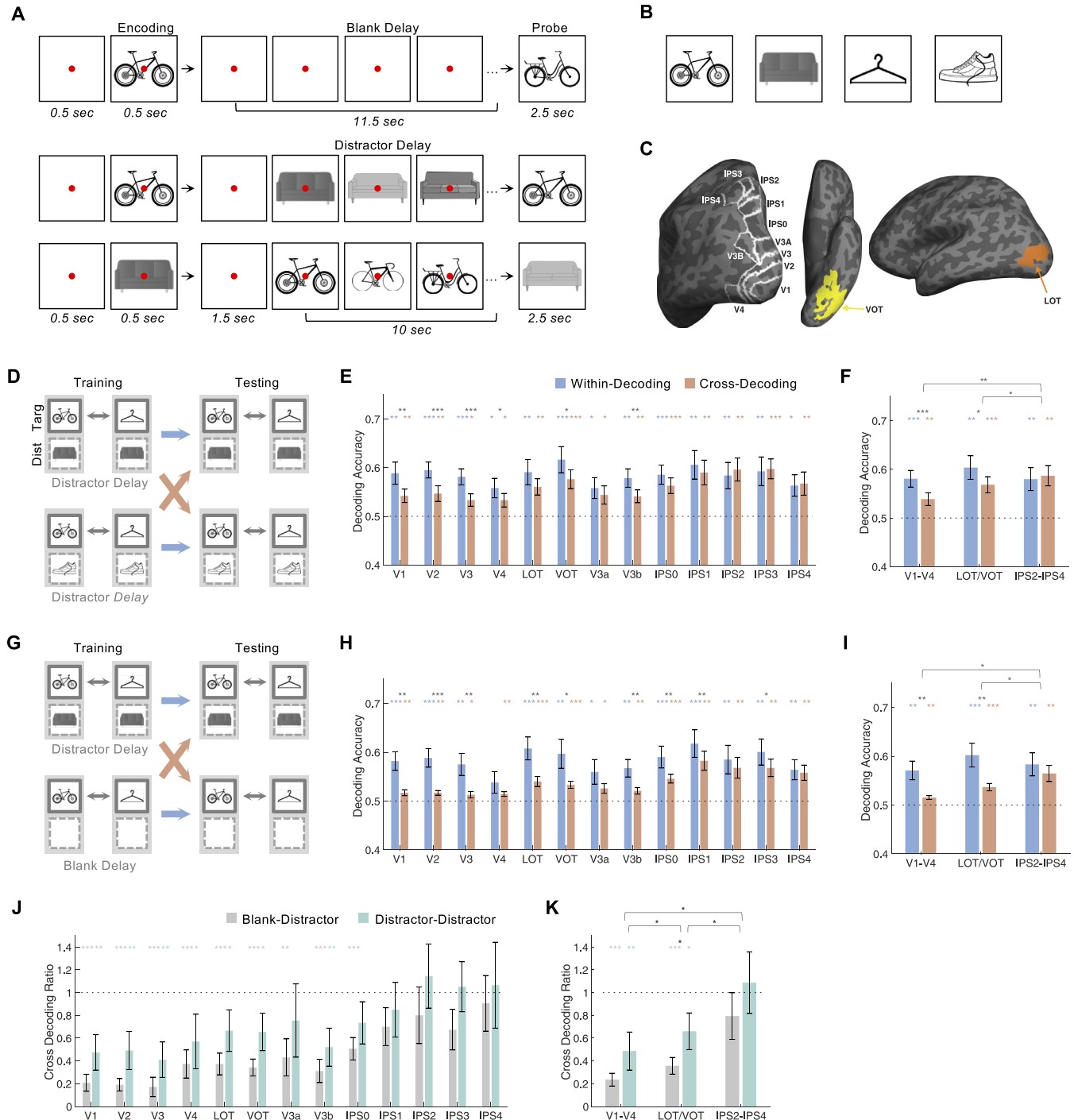

**Fig 1. Experiment 1 design and results.** (**A**) Example trials showing the trial sequence. In each trial, a single target is shown; after an extended delay period filled with either a blank screen or a sequential presentation of exemplars of another object, a probe is shown. The probe was either an exact match or a different exemplar of the same object type. Copy-free object images are used here and in subsequent figures for illustration purposes. Web-sourced photographs of real objects were used in the experiments. These images are available from the online data deposit. (**B**) The 4 types of objects used. (**C**) Example ROIs shown on the inflated cortical surfaces. (**D**) An illustration of within- and cross-target decoding between trials with different distractors. Here, a linear decoder is trained to discriminate bikes from hangers (images in the solid boxes) when either couches or shoes are the distractor (images in the dashed boxes); the decoder is then asked to discriminate bikes from hangers either under the same (within-decoding, the blue arrows) or a different distractor condition (cross-decoding, the orange arrows). (**E** and **F**) Results for all the ROIs and the 3 ROI sectors, respectively. (**G** to **I**) Same as F to H but between trials with and without distractors. (**J** and **K**) Cross-decoding ratios for both types of cross-decoding for all the ROIs and the 3 ROI sectors, respectively. In E, F, H, and J, the colored symbols above the bars mark the significance of each bar compared to chance (.5). The black symbols above the colored symbols mark the significance

of the difference between each bar pair. The black symbols above the brackets mark the significance of the difference in cross-decoding drop between pairs of ROI sectors. The horizontal dashed line indicates chance-level decoding. In J and K, the colored symbols above the bars mark the significance of the ratio compared to 1. The black symbols above the colored symbols mark the significance of the ratio difference between the 2 types of cross-decoding. The black symbols above the brackets mark the significance of the difference in the overall ratio between pairs of ROI sectors. All stats reported are corrected for multiple comparisons. Error bars indicate SE. * $p < .05$, ** $.001 < p < .01$, *** $p < .001$. Targ, target; Dist, distractor. Data are available from S1 Data and at osf.io/8rbkh/.

VWM, similar-looking exemplars of a given object were used and the probe was either the same or another exemplar of the target object. As in our previous study [5], behavioral performance did not differ between trials with and without distractors (94% correct for both; $t(13)$ = .35, $p = .73$; paired $t$ test, 2-tailed; S1B Fig).

fMRI responses were examined from regions of interest (ROIs) across OTC and PPC (Fig 1C), including early visual areas V1 to V4, object processing regions in lateral occipitotemporal (LOT) and ventral occipitotemporal (VOT) cortex, and parietal topographic areas V3a, V3b, and IPS0 to IPS4. As in a recent study [31], analysis was performed on the inflated cortical surface (vertices) rather than on the cortical volume (voxels) of each participant, as surface-based analysis has been shown to exhibit more sensitivity and better spatial selectivity ([35,36]; the average number of vertices in each ROI is reported in S2A Fig). From the averaged fMRI response time courses of each ROI (S2B Fig), a delay period before the onset of the probe was defined, and responses within this period were averaged in each vertex to generate the fMRI response pattern for each ROI in each condition (see Materials and methods). Significant VWM decoding of the target was obtained across almost all the ROIs regardless of distractor presence (see the asterisks marking the significance levels in S3A Fig; 1-sample $t$ tests, 1-tailed as only results above chance were meaningful, and corrected for the 2 tests performed in each ROI; this applies to all subsequent analysis of the same type; decoding for each time point is shown in S3C Fig). Decoding did not differ between trials with and without distractors in all the ROIs (S3A Fig; paired $t$ tests, 2-tailed), replicating prior findings when distractor presence was unpredictable [5]. To compare across the ROIs and to maximize the contrast, 3 ROI sectors were formed at the 3 ends of the ROIs with results averaged within the ROIs in each sector: a posterior sector including V1 to V4, a ventral sector including LOT and VOT, and a dorsal sector including IPS2 to IPS4. Similar decoding results were obtained in these ROI sectors as in the individual ROIs (S3B Fig). Comparison across sectors revealed no main effects of delay condition or sector, and no interaction between the two (repeated measures ANOVA, $Fs$ <1.93, $ps$ > .16; this applies to all subsequent analyses of the same type). The strengths of VWM representations were thus similar across the 3 ROI sectors and across trials with and without distractors. As before [5,6], successful decoding was also obtained for the distractors during the delay period in all the ROIs (S3C Fig).

To compare decoding accuracy across conditions assumes that decoding accuracy follows signal strength in a linear manner. Simulation results show that such a relationship indeed holds for a large part of the signal strengths, except at the very low and very high signal strengths (Supplementary Results and S4 Fig). In those cases, there exists a nonlinear relationship, resulting in either floor or ceiling decoding accuracies. Nonetheless, a probit transformation (i.e., a transformation by the inverse of the normal function) is able to remove the nonlinearity and return a mostly linear function even at the very low and very high ends of decoding (except when decoding is at chance). If we assume that fMRI noise is largely random, then a probit transformation of the decoding accuracy should allow the recovery of a more linear relationship between the decoding accuracy and the underlying signal strength, enabling more valid statistical comparisons between conditions when baseline decoding differs. Thus,

in all the decoding results reported below, stats for both the untransformed and the probit-transformed decoding accuracies were reported. Results were virtually identical with and without such transformation, if not stronger with the transformed accuracies in some cases.

To document whether an orthogonal target and distractor representational geometry in VWM is present, how well a decoder trained in one delay condition may generalize its performance to another delay condition was tested. For example, a decoder would be trained to differentiate VWM representations for a bike and a hanger when couches were the distractors. The decoder would then be asked to differentiate the same 2 target objects either under the same delay condition, or under a different delay condition with shoes being the distractors or with no distractors shown during the delay (Fig 1D and 1G). Training and testing were performed in both directions and the results were averaged.

For training and testing between delay periods with different distractors (Fig 1D), significant cross-decoding was observed in all the ROIs. However, while a large and significant cross-decoding drop was also prevalent in OTC and lower PPC regions, the drop was absent in higher PPC regions (Fig 1E; see S5A Fig for within- and cross-decoding of each time point). Similar results were observed across the 3 ROI sectors (Fig 1F). Here, while the cross-decoding drop was significant in V1-V4 and LOT/VOT, it was not in IPS2-IPS4. Comparison across the sectors revealed a main effect of cross-decoding drop ($Fs > 7.00$, $ps < .03$ for both untransformed and transformed decoding accuracies), no effect of sector ($Fs < 1.88$, $ps > .17$ for both), but an interaction between the two ($Fs > 5.78$, $ps < .01$ for both). Further pairwise comparisons revealed that cross-decoding drop was significantly greater in both V1-V4 and LOT/VOT than it was in IPS2-IPS4 ($ts > 2.22$, $ps < .034$ for both), with no difference between V1-V4 and LOT/VOT ($ts < .49$, $ps > .32$ for both; all 1-tailed and corrected for the 3 tests performed across the 3 sectors; 1-tailed as the existence of a more distractor-tolerant VWM representation in PPC than in OTC was explicitly tested based on prior research showing greater distractor resilience in VWM decoding in PPC than in EVC, see [5–7]; this applies to all subsequent tests of the same type).

Similar results were obtained for training and testing between delay periods with and without distractors (Fig 1G). Here, significant cross-decoding was observed in all the ROIs. However, while a large and significant cross-decoding drop was prevalent in OTC and lower PPC regions, the drop was minimal in higher PPC regions (Fig 1H; see S5B Fig for within- and cross-decoding of each time point). Similar results were observed across the 3 ROI sectors (Fig 1I). Comparison across the sectors revealed a main effect of cross-decoding drop ($Fs > 10.31$, $ps < .01$ for both), a main effect of sector ($Fs > 3.94$, $ps < .033$ for both), and an interaction between the two ($Fs > 4.66$, $ps < .019$ for both). Further pairwise comparisons revealed that the cross-decoding drop was significantly greater in V1-V4 and LOT/VOT than in IPS2-IPS4 ($ts > 2.02$, $ps < .048$ for both), with no difference between V1-V4 and LOT/VOT ($ts < .98$, $ps > .78$ for both).

To examine differences in the 2 types of cross-decoding and to account for decoding differences across ROIs, following our recent study documenting object representational tolerance in OTC to changes such as position and size [22], the within- and cross-decoding measures were combined to compute a cross-decoding ratio for each type of cross-decoding. This was done by first subtracting 0.5 from the within- and cross-decoding accuracies and then taking the ratio of the 2 resulting values, with a ratio of 1 indicating a complete generalization of VWM representation across delay conditions and 0 indicating no generalization. Because virtually the same results were obtained for untransformed and transformed decoding accuracies, ratios were only calculated for the untransformed decoding accuracies. The ratios were much lower than 1 in OTC and lower PPC ROIs, but no lower than 1 in higher PPC ROIs (Fig 1J and 1K). Comparisons across the sectors revealed a main effect of cross-decoding type, a main effect of sector ($Fs > 5.83$, $ps < .031$), but no interaction between the two ($F(2,26) = .042$, $p = .96$). Target cross-decoding was thus higher overall between different distractor identities than

between distractor presence and absence. Further pairwise comparisons revealed a higher overall ratio in IPS2-IPS4 than in either V1-V4 or LOT/VOT, and a higher ratio in LOT/VOT than in V1-V4 (*ts* > 2.38, *ps* < .025).

To directly visualize the representational space structure, instead of performing cross-decoding, pairwise decoding was obtained for the 4 conditions in each cross-decoding analysis to construct a representational dissimilarity matrix (RDM; see [37]; untransformed decoding accuracies were used here, as virtually the same results were obtained for the untransformed and transformed decoding accuracies). This RDM was then projected onto a 3D space using multidimensional scaling (MDS; [38]) (Fig 2A and 2E). Because the RDM only contained 4 conditions, a 3D space projection can fully capture all the representational variance. If targets and distractors are coded orthogonally, then target representations across distractor conditions should be parallel and likewise distractor representations across targets should be parallel (Fig 2B). Angles were extracted from each participant from each ROI (S6 and S7 Figs). These angles were then averaged within each ROI sector (Fig 2C left, 2D left, 2F left and 2G left). For target-target angles, across trials with different distractor conditions and across trials with and without distractors, the angles were less than 20˚ in IPS2-IPS4 but greater than 40˚ in both V1-V4 and LOT/VOT. Pairwise comparisons revealed that the angles were significantly smaller in IPS2-IPS4 than in either V1-V4 or LOT/VOT (*ts* > 4.40, *ps* < .01), with no difference between the latter two (*t(13)* < 1.26, *ps* > .22) (Fig 2C left and 2F left). For distractor-distractor angles, angles from all 3 sectors were around 20˚ or less with no pairwise differences reaching significance (*ts* < 2.47, *ps* > .25) (Fig 2D left and 2G left).

Because in 3 or higher dimensions, the angle between 2 vectors is always a positive number (as there is no mathematically unique way to determine the sign of an angle), due to the existence of variance across participants, the averaged angle would always be positive. As such, to obtain a better estimate of the true angles of the representational geometry, RDMs were first averaged for N-1 participants to generate a group-averaged RDM. Angles were then calculated from the MDS projection of this group-averaged RDM. This procedure was repeated N times with each participant serving as the left-out one to generate N angle estimates to enable statistical analyses of these angles. From these group-averaged RDMs, the average target-target angles were V1-V4: 73.0˚, LOT/VOT: 62.69˚, and IPS2-4: 0.0˚ between trials with different types of distractors, and were V1-V4: 76.28˚, LOT/VOT: 84.65˚, and IPS2-4: 0.92˚ between trials with and without distractors. The average distractor-distractor angles were V1-V4: 11.63˚, LOT/VOT: 12.97˚, and IPS2-4: 0.0˚ between trials with different types of distractors, and were V1-V4: 11.59˚, LOT/VOT: 16.52˚, and IPS2-4: 1.19˚ between trials with and without distractors. For target-target angles, pairwise comparisons revealed that the angles were significantly smaller in IPS2-IPS4 than in either V1-V4 or LOT/VOT for both types of comparisons (*ts* > 32.45, *ps* < .001), with the angles being greater in V1-V4 than LOT/VOT between trials with different types of distractors (*t(13)* = 6.39, *p* < .001), and the opposite between trials with and without distractors (*t(13)* = 3.97, *p* = .0016) (Fig 2C right and 2F right). For distractor-distractor angles, pairwise comparisons revealed for both types of comparisons that the angles were significantly smaller in IPS2-IPS4 than in either V1-V4 or LOT/VOT (*ts* > 24.73, *ps* < .001), with the angles being greater for VOT/LOT than V1-V4 (*ts* > 3.48, *p* < .01) (Fig 2D right and 2G right). Thus, only IPS2-IPS4 showed a near zero-degree angle for both, indicating parallel target coding across distractor conditions and parallel distractor coding across different targets. This directly demonstrates an orthogonal representational geometry in the target and distractor representational space and supports the interpretation of the cross-decoding results. Together, these results show that while VWM representations in OTC interact and are entangled with those of the distractors shown during the delay, in higher PPC regions there exists orthogonal target and distractor representations. This enables target representations to

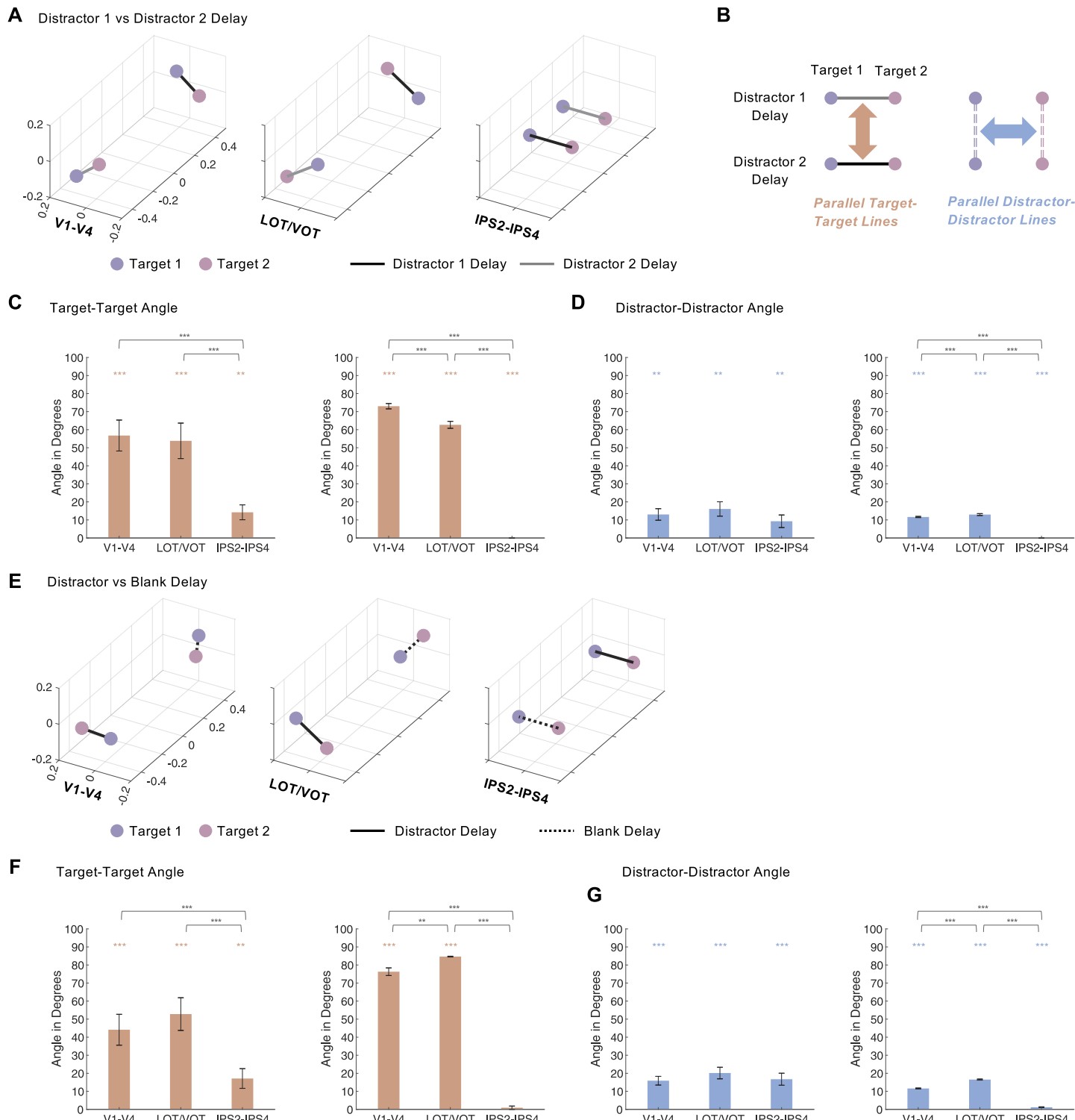

**Fig 2. Experiment 1 angles of target and distractor representations.** (**A**) Representational space for targets and distractors with different target-distractor pairing for the 3 ROI sectors across trials with different types of distractors. Each representational space geometry is an MDS projection of the grouped-averaged RDM of the 4 conditions included. Here, 2 types of targets (pink and purple) are paired with 2 types of distractors (black and gray lines). In a given trial, only 1 target (pink or purple) is shown with 1 type of distractor (black or gray line) (see Fig 1A). See main text for more details. (**B**) A schematic drawing depicting an orthogonal target-distractor representation. If orthogonality exists between target and distractor representations, then the line depicting 2 different targets in the same distractor condition (the target-target line, i.e., the gray and black lines) should be parallel across the different distractor types; likewise, the line depicting 2 different distractors in the same target condition (the distractor-distractor line, i.e., the pink and purple double dashed lines) should be parallel across the different target types. (**C**) Target-target angles and (**D**)

distractor-distractor angles across trials with different types of distractors. In each plot, angles calculated from the RDMs of the individual participants are shown on the left and those from the group RDMs are shown on the right. See main text for more details. (**E**) Representational space for targets and distractors with different target-distractor pairing for the 3 ROI sectors across trials with and without distractors. (**F**) Target-target angles and (**G**) distractor-distractor angles across trial with and without distractors. The colored symbols above the bars mark the significance of the angles above 0˚, and the black symbols above the brackets mark the significance of pairwise ROI sector comparisons. All stats reported are corrected for multiple comparisons. Error bars indicate SE. ** .001 < $p$ < .01, *** $p$ < .001. Data are available from S1 Data and at osf.io/8rbkh/.

be read out independently of those of the distractors, effectively resisting distraction. These results thus reveal for the first time the presence of a distractor-tolerant and stable VWM code in human PPC.

## Untangling target and target representations in VWM

To understand whether representation orthogonalization may be a general and effective mechanism that our brain uses to combat interference, including interference among VWM targets, Experiment 2 tested whether such a representational geometry exists for 2 concurrently held targets in VWM. In a previous study, 2 color items presented at different spatial locations were shown to form orthogonal representations in VWM in macaque LPFC, FEF, PPC, and V4 neuronal representational space [30]. This result may not be surprising given that location coding is prevalent throughout the primate brain [29], allowing items at different locations to recruit distinctive neuronal populations. However, it is unclear if such a representation can still be formed to combat interference in VWM when spatial location is no longer available for item differentiation.

To test this, across 2 scan sessions, 12 human participants (6 females) viewed a sequential presentation of 2 different target objects at fixation (a shared location) and were asked to retain the 2 targets in VWM over a long delay period (Fig 3A). The probe object shown at the end of the delay was either the same or a different exemplar of one of the target objects. The same 4 types of objects were used (Fig 3B), and the same set of ROIs were examined (see S8A and S8B Fig for the average vertices in each ROI and the fMRI response time courses for each ROI, respectively). To probe the existence of representation orthogonalization, following Experiment 1, a linear decoder was trained to classify the fMRI response patterns of a pair of target objects during the VWM delay period when each was retained with a third object (e.g., bike and couch versus hanger and couch). The same classifier was then tested in the same condition (within-decoding; bike and couch versus hanger and couch) and when each object was retained with a fourth object (cross-decoding; bike and shoe versus hanger and shoe; Fig 3B). Following the same logic as that of Experiment 1, if 2 targets retained in VWM form orthogonal representations, similar within- and cross-decoding should be obtained. However, if representations of the 2 targets entangle in VWM, then a lower cross-than-within-decoding is expected.

Behavioral performance was at 93% correct. During the encoding period, decoding of target presentation order (e.g., bike then couch versus couch then bike; S9A Fig) was above chance in all OTC areas and lower PPC areas, but not in higher PPC areas; however, during the delay period, no ROI showed above-chance decoding of presentation order (S9B Fig; see S9D Fig for presentation order decoding of each time point). Similar results were obtained for the 3 ROI sectors (S9C Fig). Comparison across the sectors revealed a main effect of decoding period ($F_{(1,11)}$ = 15.12, $p$ = .0025), an effect of sector ($F_{(2,22)}$ = 4.77, $p$ = .019), and an interaction between the two ($F_{(2,22)}$ = 20.52, $p$ < .001). Further pairwise comparisons revealed that order decoding accuracy difference between encoding and delay was greater in V1-V4 and LOT/VOT than in IPS2-IPS4 and greater in V1-V4 than in LOT/VOT ($ts$ > 1.98, $ps$ < .037).

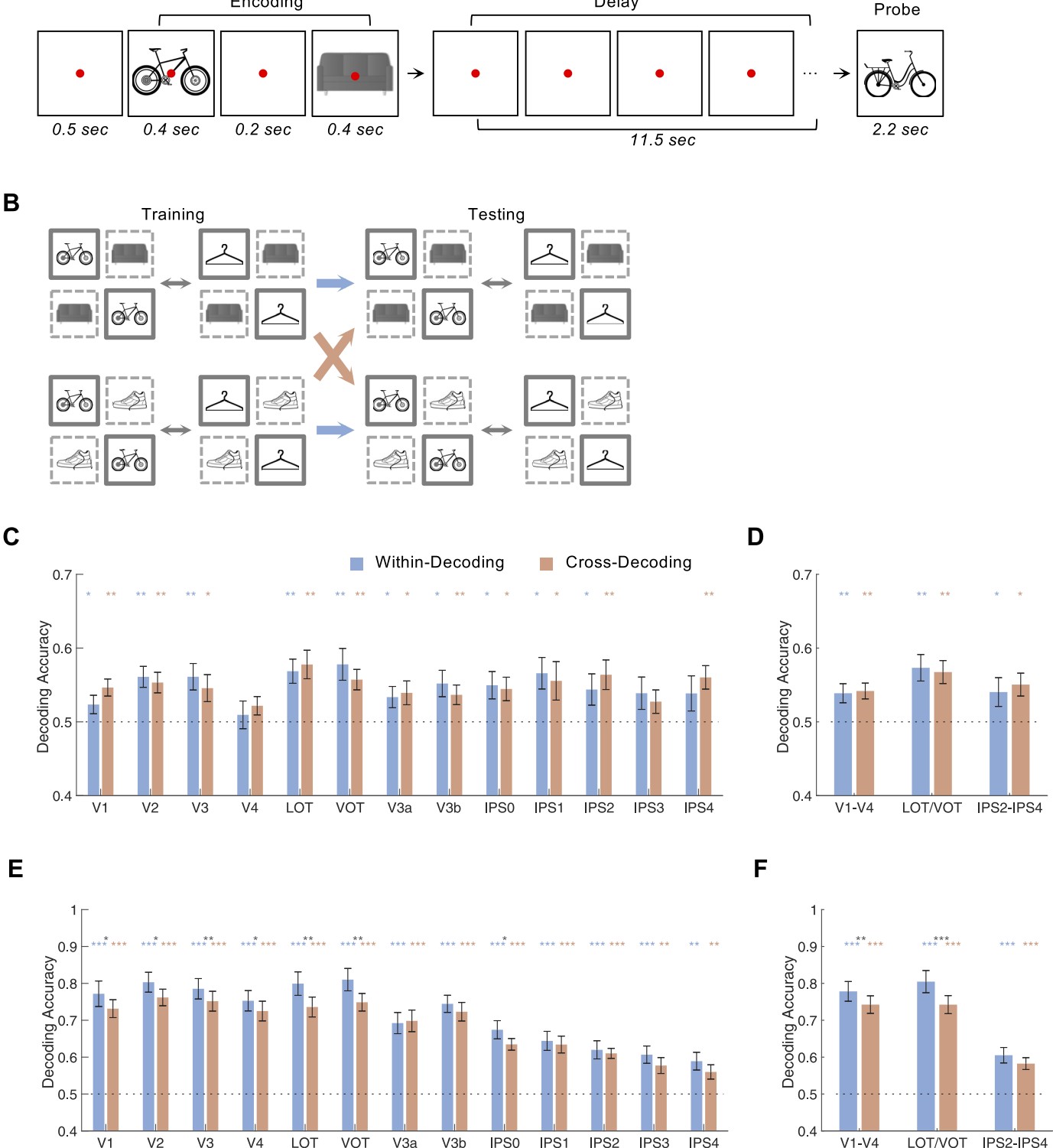

**Fig 3. Experiment 2 design and results.** (**A**) Example trials showing the trial sequence. (**B**) Illustrations of within- and cross-target decoding when a given target (images in solid boxes) is paired with different targets (images in dashed boxes). Here, a decoder is trained to discriminate bikes from hangers when each is paired with couches (top) or shoes (bottom). The trained decoder is then asked to discriminate bikes from hangers either with the second target object held constant (within-decoding, the blue arrows) or changing (cross-decoding, the orange arrows). (**C** and **D**) Results for all the ROIs and the 3 ROI sectors, respectively, during VWM delay. (**E** and **F**) Results for all the ROIs and the 3 ROIs sectors, respectively, during VWM encoding. The colored symbols above the bars mark the

significance of each bar compared to chance (.5). The black symbols above the colored symbols mark the significance of the difference between each bar pair. The horizontal dashed line indicates chance-level decoding. All stats reported are corrected for multiple comparisons. Error bars indicate SE. * $p < .05$, ** $.001 < p < .01$, *** $p < .001$. Data are available from the S1 Data and at osf.io/8rbkh/.

Given that presentation order had no measurable effect on VWM storage, to increase power, trials containing the 2 presentation orders of the same pair of objects were combined to assess VWM delay decoding of the target objects. During VWM delay, significant within-decoding and cross-decoding were found in almost all the ROIs. Importantly, there was no cross-decoding drop (Fig 3C; see S10 Fig for within- and cross-decoding of each time point for each ROI). The same results were obtained for the 3 ROI sectors (Fig 3D). Comparison across the sectors revealed a main effect of ROI ($Fs > 4.79$, $ps < .019$ for both untransformed and transformed decoding accuracies), but no effect of cross-decoding drop or an interaction between the two ($Fs < .84$, $ps > .44$ for both). It is worth noting that the overall within-VWM decoding level was similar between Experiments 1 and 2, and yet while cross-decoding was capable of dropping in Experiment 1, it did not drop in Experiment 2. In Experiment 2 during the delay period, the average cross-decoding was numerically higher than within-decoding in both V1-V4 and IPS2-4; although cross-decoding was numerically lower than within-decoding in LOT/VOT, power analysis (at a power of .80) shows that to make this difference significant (1-tailed), we will need 449 participants. Meanwhile, both within- and cross-decoding were significantly above chance in all 3 ROI sectors. Thus, the lack of a cross-decoding drop in all 3 ROI sectors did not appear to be due to a lack of power; rather, it reflected a similar level of within- and cross-decoding performance. Overall, these results show that the representation of a target in VWM is not affected by another concurrently held target in VWM, indicating orthogonal representations of the 2 targets in VWM in both OTC and PPC.

Using the same approach, whether a similar orthogonal target representation is present during VWM encoding was also examined. While significant within-decoding and cross-decoding were found in all the ROIs, all OTC areas also showed a significant cross-decoding drop; such a drop, however, was absent in the majority of the PPC areas (Fig 3E). Similar results were found for the 3 ROI sectors, with significant within- and cross-decoding in all three, a significant cross-decoding drop in V1-V4 and LOT/VOT, but not in IPS2-IPS4 (Fig 3F). Comparison across the sectors revealed a main effect of cross-decoding drop ($Fs > 16.61$, $ps < .01$ for both untransformed and transformed decoding accuracies), an effect of sector ($Fs > 43.57$, $ps < .001$ for both), and an effect of interaction between the two (untransformed, $F(1,11) = 3.26$, $p = .057$; transformed, $F(1,11) = 5.52$, $p = .011$). Pairwise comparisons revealed that the difference in cross-decoding drop was greater in LOT/VOT than in IPS2-IPS4 (untransformed, $t(11) = 2.33$, $p = .060$; transformed, $t(11) = 3.23$, $p = .012$), slightly greater in V1-V4 than in IPS2-IPS4 (untransformed, $t(11) = 1.03$, $p = .16$; transformed $t(11) = 1.81$, $p = .073$), and slightly greater in LOT/VOT than in V1-V4 (untransformed $t(11) = 1.54$, $p = .11$; transformed $t(11) = 1.65$, $p = .063$). These results show that during VWM encoding, an object's representation is modulated by the identity of another encoded object in OTC, indicating entangled target representations. The same effect appears to be weaker in PPC.

Within each ROI sector, a direct comparison of the effect of decoding (within versus cross) and VWM processing stage (encoding versus delay) revealed that, in V1-V4, there was an effect of decoding (untransformed, $F(1,11) = 4.35$, $p = .061$; transformed, $F(1,11) = 6.63$, $p = .026$), an effect of processing stage ($Fs > 53.69$, $ps < .001$ for both), and an effect of interaction (untransformed, $F(1,11) = 3.42$, $p = .091$; transformed $F(1,11) = 5.23$, $p = .043$). In LOT/VOT, all 3 effects were significant ($Fs > 5.63$, $ps < .037$ for both). In IPS2-IPS4, there was no effect of decoding ($Fs < .28$, $ps > .60$ for both), an effect of processing stage ($Fs > 10.54$, $ps < .01$ for

both), and a marginal effect of interaction (untransformed, $F(1,11) = 4.38$, $p = .060$; transformed, $F(1,11) = 4.69$, $p = .053$). These results show that cross-decoding was greater during VWM encoding than delay in OTC; the same effect appears to be weaker in PPC.

To directly visualize the target-target representational space during VWM delay and encoding periods, as in Experiment 1, RDMs were constructed from the 4 conditions involved in each cross-decoding analysis, and the averaged RDM results were projected into 3D spaces using MDS for each ROI and ROI sector (S11A and S12A Figs for each ROI and Fig 4A and 4C for each ROI sector; untransformed decoding accuracies were used, as results were similar across untransformed and transformed decoding accuracies). Angles of target-target representations were then measured from these representational spaces (S11B and S12B Figs for each ROI and Fig 4B and 4D for each ROI sector). During the delay period, the averaged angles calculated from the individual participants were below 20° for all 3 sectors (Fig 4B, left), with no difference in pairwise sector comparisons ($ts < .71$, $ps > .90$). The averaged angles calculated from the group-averaged RDMs were .74°, 0.0° and 0.0°, for V1-V4, LOT/VOT and IPS2-4, respectively (Fig 4B, right), again with no difference in pairwise sector comparisons ($ts < 1.00$, $ps > .50$). During the encoding period, the averaged angles calculated from the individual participants were around 20° for IPS2-4, but greater for V1-V4 and LOT/VOT (Fig 4D, left). The angle was smaller in IPS2-4 than in the other 2 sectors (compared to V1-V4, $t(11) = 2.13$, $p = .084$; compared to LOT/VOT, $t(11) = 2.96$, $p = .039$; with no difference between V1-V4 and LOT/VOT, $t(11) = 1.53$, $p = .16$). The averaged angles calculated from the group-averaged RDMs were 39.26°, 55.02°, and 8.81° for V1-V4, LOT/VOT, and IPS2-4, respectively (Fig 4D, right). Here, the angles were significantly smaller in IPS2-IPS4 than in the other 2 sectors and significantly smaller in V1-V4 than in LOT/VOT ($ts > 10.83$, $ps < .001$).

ANOVA between VWM processing stage (encoding versus delay) and pairs of ROI sectors revealed strong interaction in the group RDM angles and moderate interaction in the individual RDM angles, indicating a smaller angle difference between VWM encoding and delay in IPS2-IPS4 than in the other sectors (between V1-V4 and IPS2-IPS4, grouped RDM, $F(1,11) = 99.49$, $p < .001$, and individual RDM, $F(1,11) = 1.08$, $p = .32$; and between LOT/VOT and IPS2-IPS4, grouped RDM, $F(1,11) = 318.19$, $p < .001$, and individual RDM, $F(1,11) = 3.87$, $p = .075$). These results show that during VWM delay an orthogonal representational geometry exists for target-target representation across all 3 ROI sectors; during VWM encoding, however, while the target-target representational space was still near orthogonal in IPS2-IPS4, it was far from being orthogonal in V1-V4 and LOT/VOT. These results support the interpretation of the cross-decoding results.

Overall, the results show that target representations are entangled during VWM encoding in OTC but less so in PPC. However, such entanglement appears to dissipate over the course of VWM retention, with both OTC and PPC forming orthogonal representations for the 2 targets stored in VWM. This shows that representation orthogonalization is likely a general and effective mechanism that our brain uses to combat interference, including interference among the different VWM targets.

## Discussion

Using fMRI pattern cross-decoding and direct angle measurements of the representational space, this study unveils a common representational geometry that our brain uses to combat distractions in VWM. Across 2 experiments, the human PPC is shown to orthogonalize the representations of the different streams of information that are coded concurrently, whether they are targets and distractors, or different targets retained together in VWM. The latter is also seen in OTC. Such a representational geometry enables independent information readout,

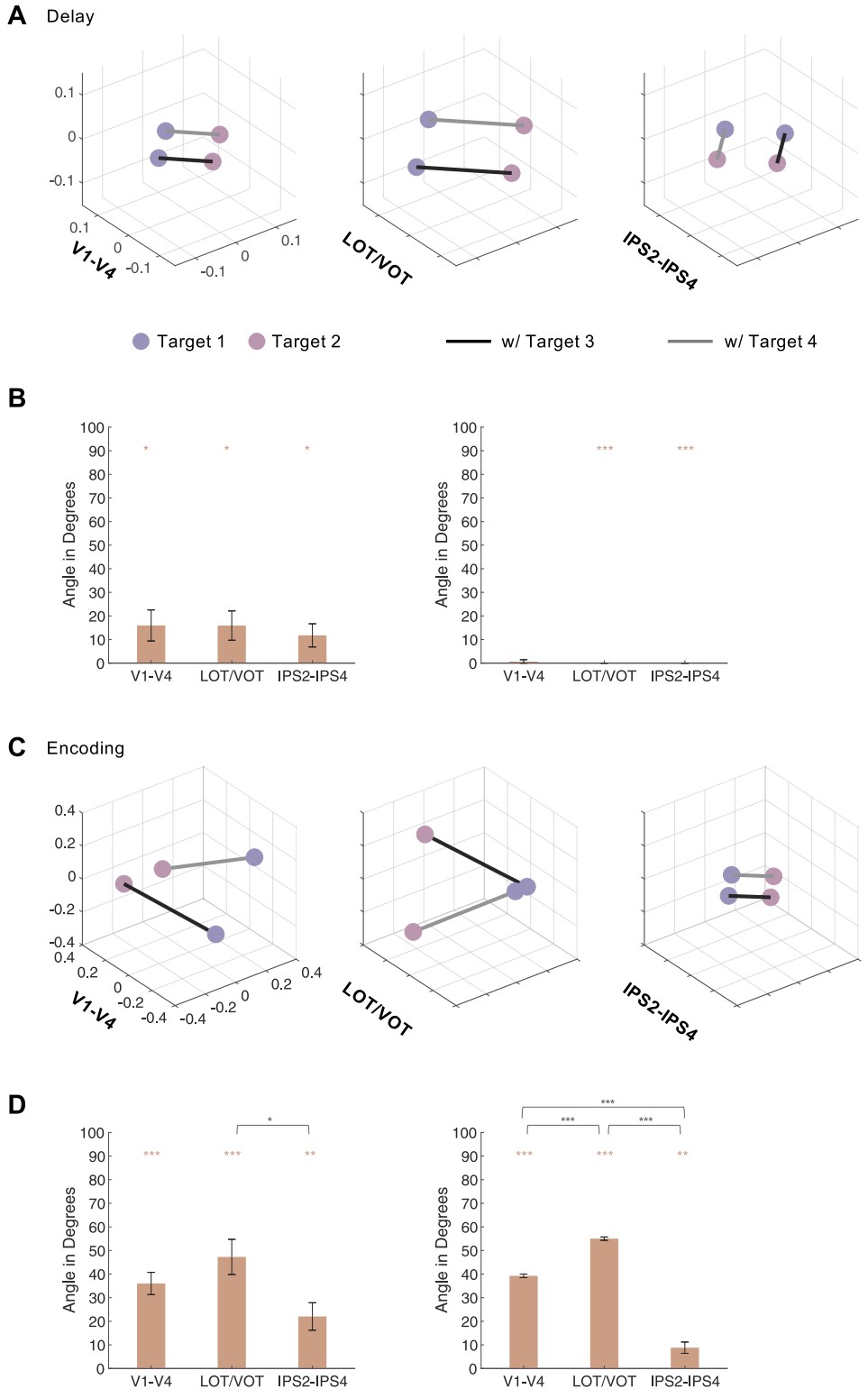

**Fig 4. Experiment 2 angles of target-target representations.** (A) Representational space for targets for the 3 ROI sectors during VWM delay. Each representational space geometry is an MDS projection of the grouped-averaged RDM of the 4 conditions included. Here, 2 types of targets (pink and purple) are paired with 2 other types of target (black and gray lines). In a given trial, 1 target (pink or purple) is shown with another target (black or gray line) (see Fig 2A). If orthogonality exists between target and target representations, then the line depicting 2 different targets

(pink and purple) paired with the same target in a trial (the gray line) should be parallel when these targets are paired with a different target (the black line). See main text for more details. (**B**) Target-target angles during VWM delay. In each plot, angles calculated from the RDMs of the individual participants are shown on the left and those from the group RDMs are shown on the right. See main text for more details. (**C**) Representational space for targets for the 3 ROI sectors during VWM encoding. (**D**) Target-target angles during VWM encoding. The colored symbols above the bars mark the significance of the angles above 0°, and the black symbols above the brackets mark the significance of pairwise ROI sector comparisons. All stats reported are corrected for multiple comparisons. Error bars indicate SE. * $p$ < .05, ** .001 < $p$ < .01, *** $p$ < .001. Data are available from S1 Data and at osf.io/8rbkh/.

effectively preventing distraction from other concurrently coded information. It is an adaptive representational scheme as it enables PPC to maintain the stability and invariance of its VWM representations while at the same time accommodating the representations of distractors and other VWM targets. This can be highly advantageous in situations where distracting information becomes task-relevant and in dual-task settings where distractor processing is actually task-relevant (they are no longer distractors in this regard).

In Experiment 1, a continuous stream of highly salient distractors was shown during the VWM delay period, making them difficult to suppress or block. VWM target representations in OTC were found to be entangled with those of the distractors shown during the delay, with target representations being different when distractors were present and when they were absent, and different when different distractors were shown. In contrast, targets and distractors were untangled and formed orthogonal representations in higher PPC regions, allowing target representations to be minimally impacted by the distractors and exhibiting tolerance to the presence and the identities of the distractors. Such a representational geometry presents a simple and elegant solution to combat distraction without relying on an active suppression or blocking mechanism that has traditionally been associated with distractor filtering in other situations [8,9]. The utilization of an orthogonal representational geometry in PPC but not in OTC may provide a mechanistic account for why distractor resilience is found to be greater in PPC than in OTC [5–7].

It is worth noting that in OTC, distractor representation was much greater than target representation during VWM encoding, but in PPC, a similar level of representation was seen (S3C Fig). Thus, an active suppression/filtering mechanism could have partially contributed to distractor suppression in PPC. This would echo prior work showing that visual representations in PPC are more task-driven than those in OTC [25,26,28]. Nonetheless, task-irrelevant sensory information could still be represented robustly in PPC [26]. In the present study, despite distractor representation in PPC being on par with target representation during VWM encoding, it was still much higher than target representation during VWM delay. An active suppression/filtering mechanism, even if it were deployed, was thus unable to completely suppress or remove distractor representation. To combat this, the present results showed that PPC, but not OTC, used an orthogonal representational scheme to separate target and distractor representations. PPC may thus use both a suppression/filtering mechanism and representation orthogonalization to combat distraction in VWM, with the latter mechanism being potentially more adaptive as it can accommodate the simultaneous representations of both targets and distractors in VWM.

Using colored moving dot displays in a perceptual decision-making task, a recent study reported that the coherence levels of the target feature (color) and the coherence levels of the distractor feature (motion) form orthogonal axes in the EVC and intra-parietal sulcus feature representational space but parallel axes in midvisual areas and superior parietal lobule [39]. However, such a result could be driven by feature-dependent versus feature-independent coding of coherence levels, rather than a difference between target and distractor per se, as control

conditions examining feature coding without specific target and distractor assignment were not included to rule out this alternative account.

OTC is best known for its ability to form transformation-tolerant object representations, enabling us to recognize an object despite changes in viewing conditions. This is achieved by untangling object identity and nonidentity features in EVC to form largely orthogonal representations of these features in higher ventral visual areas [17–22]. Such an ability is regarded as a defining feature of primate high-level vision [14,15]. The present study shows that a similar untangling process is occurring in PPC between target and distractors in VWM. A defining feature of dorsal visual processing may thus lie in the formation of distractor-tolerant object representations during goal-direct visual processing, consistent with its overall role in adaptive visual processing [40,41].

An orthogonal target-distractor representational scheme is agnostic to the exact neuronal populations involved: It may be implemented by distinctive neuronal subpopulations representing different streams of information, or within the entire neuronal population to take full advantage of the available representational resources. In the latter case, as long as target representations remain orthogonal to those of the distraction conditions, target information can still be read out independently of distraction by downstream regions. With fMRI pattern responses summing over large neuronal populations, both scenarios would produce the same results. Further research involving direct neuronal recording is thus needed to distinguish between these 2 representational possibilities.

Using the same approach as Experiment 1, Experiment 2 shows that the same representational geometry also applies when 2 targets are held together in VWM. Here again, there are 2 types of concurrently coded visual information that need to be kept apart in VWM. In this case, because the 2 targets were shown sequentially at the same spatial location, location-based encoding involving distinctive neuronal populations can no longer be used to segregate representations [30]. By applying an orthogonalization representational scheme, however, the 2 targets could still be represented independently without interfering with each other in VWM. This shows that representation orthogonalization is likely a general and effective mechanism that our brain uses to combat interference in VWM, whether between targets and distractors or between different targets held together in VWM. In Experiment 2, while orthogonal target representations were present in both OTC and PPC during VWM delay, entangled representations were found in OTC during VWM encoding (with such entanglement being weaker in PPC). This shows that even though the 2 targets were entangled in OTC during encoding, they became untangled and orthogonalized during delay. This could be part of the VWM consolidation process driven by feedback signals from association areas such as PFC and PPC to OTC. Indeed, feedback has been shown to sustain and restore VWM representations after distraction in sensory regions [34] and drive the representational content of VWM in OTC [31]. Given PPC's ability to form orthogonal target and distractor representations as shown in Experiment 1, feedback from association areas such as PPC likely helps untangle the encoded target representations in OTC, resulting in orthogonal representations in both during VWM delay. Further work with better temporal resolution is needed to test this idea and to provide us with a more detailed understanding of the untangling process.

Previous studies have shown that information about the target presentation order is retained during VWM delay when such information is critical to the VWM task [42,43]. In Experiment 2, while the target presentation order was decodable during VWM encoding in V1-V4 and LOT/VOT, it was not in IPS2-IPS4. During VWM delay, the target presentation order was not decodable in any of the ROIs examined. An effect of presentation order in sensory areas during encoding may not be surprising due to the temporal separation of the presentations of the 2 target objects. The lack of this effect in higher PPC areas could be due to the

longer intrinsic time courses of associative areas [44], task irrelevance of the target presentation order, or both. It is interesting that during the delay period, an order effect was absent across OTC and PPC, likely due to such information being task-irrelevant. Thus, order may or may not be represented in VWM depending on its task relevance. Future research is needed to verify this prediction.

To understand why VWM is capacity limited [45–47], based on neurophysiological evidence of synchrony (e.g., [32,48–51]), it has been proposed that synchronized neural firing of different local neuronal assemblies, each coding a different item, coupled with long-range synchrony across associative and sensory regions, supports the retention of multiple items in VWM while reducing their mutual interference [52,53]. Under this framework, VWM capacity limitation is believed to reflect the number of items that can be successfully retained together via neural synchrony. Showing the existence of such synchrony experimentally, however, has been difficult as it requires simultaneous recording of a large population of neurons and determining their connections and correlations [54,55]. It is also unclear if the same neuronal assembly supports the representation of a target object when it is stored with different target objects in VWM; the stability in target representation would be essential if target representation is to be invariant to other concurrently stored objects. While neural synchrony involving distinctive neuronal assemblies can result in representation orthogonalization, as discussed earlier, an orthogonal representational scheme may be implemented within the same neuronal population so long as an orthogonal representational geometry can be formed. Under this representational framework, what determines VWM storage capacity would be the number of orthogonal representations that can be sustained over time, with long-range synchrony across regions potentially driving and sustaining such representations across regions [56–58]. This prediction is potentially easier to test experimentally than those involving neural synchrony. It would be exciting for future neurophysiological research to verify the existence of such a representational scheme and understand its role in determining VWM capacity.

As mentioned earlier, representation orthogonalization is employed by OTC to separate object-identity and nonidentity features during visual processing. Beyond OTC and object perception, representation orthogonalization has been shown to code targets with different priorities during VWM delay [42,43,59–61]; attention-related, but not VWM-related, processing [62]; the separation of sensory input and stored long-term memory information [63]; and the interaction between attention and VWM [30]. Representation orthogonalization is thus likely a fundamental neural mechanism that our brain uses to accommodate the processing of multiple streams of information associated with different behavioral goals.

In summary, using fMRI pattern decoding, the present study reveals that representation orthogonalization is a common representational geometry that our brain employs to combat distraction from the coding of distractors and other concurrently held targets in VWM. It is an adaptive representational scheme that departs from active suppression/filtering mechanisms traditionally associated with distractor resistance. It enables the maintenance of stable and invariant VWM representations while simultaneously accommodating the representations of distractors and other VWM targets. Such a mechanism likely plays a significant role in supporting the adaptative nature of PPC visual processing.

## Materials and methods

### Participants

Fourteen (9 females) and 12 (6 females) healthy human participants took part in Experiments 1 and 2, respectively. Four of the participants (3 females) took part in both experiments. All participants had normal or corrected-to-normal visual acuity, were all right-handed, and were

aged between 18 and 35. All participants gave their written informed consent prior to each experiment and received payment for their participation. The study was approved by the Committee on the Use of Human Subjects at Yale University (approved protocol # 0307025445). The study has been conducted in accordance with the principles of the Declaration of Helsinki. Three and 1 additional participants also took part in Experiments 1 and 2, respectively, but did not complete the second of the 2 testing sessions of each experiment. Their data were not included in the analysis.

Our sample sizes for both experiments are higher than or similar to a number of recently published VWM studies (e.g., $Ns$ = 6 and 7 for the 2 experiments in [6]; $N$ = 13, in [64]; $N$ = 16 in [65]; $N$ = 7 in [66]; $Ns$ = 13 and 14 for the 2 experiments in [67]; $N$ = 11 for [68]; and $N$ = 14, [69]).

## Main VWM experiment

Both experiments used the same 4 types of objects, and they were bikes, couches, coat hangers, and shoes (sneakers) (Fig 1B; some of these objects were used in [70–72]). To increase task difficulty and ensure that a visual code was used to retain VWM representation, similar-looking exemplars of a given object were used. All images were placed on a white square (subtended 9.73˚ × 9.73˚) and shown on a larger gray background.

In Experiment 1, each VWM trial contained a central presentation of a target object, a prolonged delay, and a probe object (Fig 1A). The probe object was either an exact match to one of the target images or a different exemplar of the same type of object. Each trial was 15 s long, with the timing of the different events as follows: fixation (.5 s)—in the form of a looming red dot alerting the participants to the imminent presentation of the target images, target image (.5 s), blank delay with a red fixation dot (1.5 s), blank delay or distractor delay with a red fixation dot (10 s), and probe image (2.5 s). In trials with a distractor delay, 20 distractor images were shown, with the 10 exemplars from the distractor object each shown twice with no back-to-back repetition of the same exemplar. Each distractor image was shown for .3 s, followed by a .2-s blank screen before the next distractor image was shown. There were a total of 16 unique trials in each run, including 12 trials for all the target and distractor object combinations (4 target object types × 3 distractor object types) and 4 trials in which no distractors were shown with 1 for each target object type. Each run started and ended with an 8-s blank period with a blue fixation dot. Successive VWM trials were sandwiched by a blank period with a blue fixation dot. Of the 15 such intertrial blank periods, three were 8 s long and 12 were 2 s long, and they were randomly distributed. In a first scan session, 13 or 14 runs of data were collected from each participant, with each run lasting 5 min 4 s; in a second scan session, 14 runs of data were collected from each participant.

Experiment 2 used a similar design to that of Experiment 1 with the following differences. In 12 of the 16 trials, 2 targets were shown sequentially. All possible 2-target combinations from the 4 types of objects and the 2 presentation orders, totaling 12 different conditions, were each shown once. In the remaining 4 trials, a single target was shown, with each of the 4 object types shown once. Data from the single-target trials were not analyzed for the purpose of the present study. Each trial was 15.2 s long, with the timing of the different events as follows (Fig 3A): fixation (.5 s)—in the form of a looming red dot alerting the participants to the imminent presentation of the target images, first target image (.4 s), blank delay with a red fixation dot (.2 s), second target image (.4 s), blank delay with a red fixation dot (11.5 s), and probe image (2.2 s). In single-target trials, the first and second target images were identical. Each run lasted 5 min 12 s. Fourteen runs of data were collected

from each participant in each of the 2 scan sessions. Other aspects of the design were identical to those of Experiment 1.

## Localizer experiments

**Topographic visual regions.** These regions were mapped with flashing checkerboards using standard techniques [73,74] with parameters optimized following [74] to reveal maps in the parietal cortex. Specifically, a polar angle wedge with an arc of 72˚ swept across the entire screen (19.07 × 13.54˚ of visual angle). The wedge had a sweep period of 32 s, flashed at 4 Hz, and swept for 8 cycles in each run (for more details, see [74]). Participants completed 4 to 6 runs, each lasting 4 min 36 s. All participants were asked to detect a dimming that could occur anywhere within the polar angle wedge, thereby ensuring attention to the whole wedge.

## Lateral and ventral occipitotemporal regions (VOT and LOT)

To identify these ROIs, following [75] and as in our previous studies [19,20,23,26,28,70], participants viewed blocks of objects and scrambled objects (all subtended approximately 9.73˚ × 9.73˚). The images were photographs of gray-scaled common objects (e.g., cars, tools, and chairs) and phase-scrambled versions of these objects. Participants monitored a slight spatial jitter, which occurred randomly once in every 10 images. Each run contained 4 blocks of each of objects, phase-scrambled objects, and 2 other conditions that were used to define another brain region. Each block lasted 16 s and contained 20 unique images, with each appearing for 750 ms and followed by a 50-ms blank display. Besides the stimulus blocks, 8-s fixation blocks were included at the beginning, middle, and end of each run. Each participant was tested with 2 runs, each lasting 4 min 40 s.

## MRI method

Each participant completed 2 sessions (1.5 h) for each experiment and a localizer session (1.5 h) containing topographic mapping and functional localizers. MRI data were collected using a Siemens Prisma 3T scanner, with a 32-channel receiver array head coil. Participants lay on their backs inside the scanner and viewed the back-projected display through an angled mirror mounted inside the head coil. The display was projected using an LCD projector at a refresh rate of 60 Hz and a spatial resolution of 1,280 × 1,024. An Apple Macbook Pro laptop was used to create the stimuli and collect the motor responses. Stimuli were created using Matlab and Psychtoolbox ([76]).

A high-resolution T1-weighted structural image (0.8 × 0.8 × 0.8 mm) was obtained from each participant for surface reconstruction. All blood-oxygen-level-dependent (BOLD) data were collected via a $T2^*$-weighted echo-planar imaging pulse sequence that employed multi-band RF pulses and simultaneous multislice (SMS) acquisition. For both the 2 main experiments and the localizers, 72 axial slices (2 mm isotropic), 0 skip, covering the entire brain were collected (TR = 800 ms, TE = 37 ms, flip angle = 52˚, SMS factor = 8).

## Data analyses

FMRI data were analyzed using FreeSurfer (surfer.nmr.mgh.harvard.edu), FsFast [77], and in-house MATLAB codes. LibSVM software [78] was used for the MVPA support vector machine analysis. FMRI data preprocessing included 3D motion correction and linear and quadratic trend removal. After reconstructing the inflated 3D cortical surface of each participant using the high-resolution anatomical data, the fMRI data from that participant were projected onto their native cortical surface. As was done in a recent study [31], all fMRI responses were

analyzed directly on the inflated cortical surface (vertices) rather than on the cortical volume (voxels) of each participant, including ROI definition and the main VWM analysis, as surface-based analysis has been shown to exhibit more sensitivity and better spatial selectivity [36,37]. The average number of vertices in each ROI are reported in S2A and S8A Figs for Experiments 1 and 2, respectively.

### ROI definitions

Following the detailed procedures described in [74] and as was done in our prior publications [5,19,20,26,79,80], by examining phase reversals in the polar angle maps, areas V1 to V4, V3a, V3b, and IPS0 to IPS4 in each participant were identified (Fig 1C). Following [75] and as was done in our prior studies [19,20,23,26,28,70], LOT and VOT were then defined as a cluster of continuous voxels in the lateral and ventral occipital cortex, respectively, that responded more to the intact than to the scrambled object images (Fig 1C). LOT and VOT loosely correspond to the location of LO and pFs [75,81,82] but extend further into the temporal cortex in an effort to capture the continuous activations often seen extending into the ventral temporal cortex.

### VWM decoding analysis

With the length of our VWM trials being 15 s for Experiment 1 and 15.2 s for Experiment 2, in each experiment, for each surface vertex, the fMRI response amplitude at each TR from the onset of the trial up to 24 s was estimated, totaling 30 TRs (with each TR being 800 ms). This was done separately for the trials in each of the 16 conditions. To obtain these estimates, 30 finite impulse response functions (FIRs) corresponding to each TR of the trial for each condition were first constructed. As each condition appeared only once per run, given the short intertrial interval (mostly 2 s) and the lag in hemodynamic responses, it was not possible to accurately estimate the amplitudes of the FIRs in each run. To overcome this, following the procedure developed in a recent study [31], 5 or 6 runs were combined together, as detailed below, before applying GLM modeling to derive the beta weight estimate for each of the 480 FIR functions (30 TRs × 16 conditions). Because the trial onset times were jittered with respect to the TR onsets in Experiment 1, trial onset times were rounded to the nearest TR before GLM modeling. This was not needed in Experiment 2 as the trial onset times coincided with the onset of the TRs. To obtain independent training and testing data for pattern decoding, runs were split into odd and even halves in each of the 2 scan sessions, with each split containing 6 or 7 runs depending on the total number of runs acquired in a given session. A GLM was then applied to each 6-run combination if the split contained 7 runs and to each 5-run combination plus a combination including all 6 runs if the split contained 6 runs. This resulted in 7 beta weight estimates for each FIR function in each split for each surface vertex. The beta weights of all the vertices in a given ROI formed our fMRI response pattern for that ROI. For each ROI, there were thus a total of 28 patterns for each TR and each condition across the 4 data splits. Note that within a split, the 7 patterns were not independent of each other as they were estimated from shared runs; however, the patterns from the different splits came from independent runs and were thus independent of each other.

 To generate a response amplitude time course from each ROI for each condition, all the beta weights across all the surface vertices within an ROI and from all 28 patterns were averaged together. The results are shown in S2B and S8B Figs for Experiments 1 and 2, respectively. Based on the peak responses from all the ROIs and conditions, in Experiment 1, the delay period was defined to be from 9.6 to 12 s before the onset of the probe at 12.5 s. In Experiment 2, due to the longer target presentation time (to accommodate the sequential presentations of

2 targets), the delay period was defined to be from 10.4 to 12.8 s before the onset of the probe at 13 s. To compare and contrast delay and encoding responses in Experiment 2, an encoding period was defined in this experiment to be from 4.8 to 8 s. The beta weights within each period were then averaged to generate an average response for that period.

Prior to our decoding analysis, to remove amplitude differences across categories, ROIs and VWM processing stages, following our previous studies [5,19,20,26,31], each fMRI response pattern was z-normalized. For a given ROI and for a pair of conditions, SVM was used for pattern decoding (LibSVM; [78]). A decoder was trained using all the response patterns from 3 splits of the data (totaling 21 patterns) to test its performance on the left-out data split (7 patterns). Training and testing were thus done on independent data sets. The training and testing order was rotated, with each data split serving as the test split and the remaining three as the training splits. Results were averaged across the 4 rotations.

Training and testing were performed separately for each pair of conditions of interest, and the decoding results were averaged across all relevant pairs of conditions to derive the average decoding performance for a given analysis. To directly compare the different ROIs, to maximize the contrast, and to streamline the analysis, based on the anatomical locations, 3 ROI sectors were formed at the 3 ends of the ROIs, with decoding performance averaged within the ROIs in each sector: a posterior sector including lower visual areas V1-V4, a ventral sector including object areas LOT and VOT, and a dorsal sector including higher PPC areas IPS2-IPS4. Within each ROI sector, decoding within each ROI was first performed and then averaged, rather than with decoding performed on the combined ROI containing the individual ROIs. This was done to allow equal weighing of the results from the different ROIs: As ROI size differed across ROIs and participants, decoding based on a combined ROI could bias the results towards the larger ROIs, which could further differ across participants. Our main comparisons focused on the differences among the 3 ROI sectors.

In Experiment 1, to compare target decoding between trials with and without distractors, target object decoding during the delay period was first examined with the distractor condition kept constant. This was done by comparing the decoding of a pair of target objects when no distractors were present and when distractors were present during the delay period (see S3A and S3B Fig; see S3C Fig for the decoding of each time point). To understand the extent to which VWM representations were impacted by the presence of distraction, a decoder was trained to decode a pair of target objects in one delay condition and its decoding performance was then tested for the same pair of target objects both in the same and in a different delay condition. For example, a decoder would be trained to differentiate VWM representations for a bike and a hanger when couches were the distractors. The decoder would then be asked to differentiate the same 2 target objects either under the same delay condition, or under a different delay condition with shoes being the distractors or with no distractors shown during the delay. Training and testing were performed in both directions, and the results were averaged. Cross-decoding was conducted for a pair of target objects across trials with different distractors shown, and across the distractor presence and absence trials (Fig 1D and 1G). The results are shown in Fig 1E, 1F, 1H and 1I (see S5 Fig for the decoding of each time point).

In Experiment 1, to examine differences in the 2 types of cross-decoding and to account for decoding performance differences across the different ROIs, following our study documenting tolerance of visual object representations in OTC to changes such as position and size [22], the within- and cross-decoding measures were combined by computing a cross-decoding ratio for the 2 types of cross-decoding. This was done by first subtracting 0.5 from the within- and cross-decoding accuracies and then taking the ratio of the 2 resulting values. A ratio of 1 would indicate equally good decoding performance within and across delay conditions and a complete generalization of VWM representation across the different delay conditions, whereas

a ratio of 0 would indicate a complete failure of such generalization. The results are shown in Fig 1J and 1K.

In Experiment 2, to examine the effect of presentation order, decoding was performed on trials containing the same 2 target objects but with different presentation orders during encoding. The effect of presentation order was examined at both VWM encoding and delay periods, and the results are reported in S9 Fig. To assess whether the representation of a given target in VWM is impacted by the concurrent retention of another target, similar to Experiment 1, a decoder was trained to decode the fMRI response patterns of a pair of target objects when each was retained with a third object (e.g., bike and couch versus hanger and couch), and its decoding performance was then tested both in the same condition (within-decoding; bike and couch versus hanger and couch) and when each object was retained with a fourth object (cross-decoding; bike and shoe versus hanger and shoe; Fig 3B). This decoding was performed during both the VWM delay period and the VWM encoding period (Fig 3C to 3F; see S10 Fig for the decoding of each time point).

## Angle calculation in VWM representational space

To directly visualize the representational space structure, instead of performing cross-decoding, pairwise decoding was obtained for the 4 conditions in each cross-decoding analysis to construct an RDM (see [38]; untransformed decoding accuracies were used here, as virtually the same results were obtained for untransformed and transformed decoding accuracies). This RDM was then projected onto a 3D space using MDS [39]. Because the RDM only contained 4 conditions, a 3D space projection can fully capture all the representational variance. If targets and distractors are coded orthogonally in VWM in Experiment 1, then target representations across distractor conditions should be parallel, and likewise, distractor representations across targets should be parallel (Fig 2B). Similarly, in Experiment 2, if 2 targets are coded orthogonally in VWM during the delay period, then parallel target representations are expected. To quantify these predictions, target-target (Experiments 1 and 2) and target-distractor (Experiment 1) angles were extracted from each participant from each ROI. These angles were then averaged within each ROI sector.

Because in 3 or higher dimensions, the angle between 2 vectors is always positive (as there is no mathematically unique way to determine the sign of an angle), due to the presence of variance across participants, the averaged angle would always be positive. As such, to obtain a better estimate of the true angles of the representational geometry, RDMs were averaged across N-1 participants to generate a group-averaged RDM. Angles were then calculated from the MDS projection of this RDM. This procedure was repeated N times with each participant serving as the left-out one to generate N angle estimates to enable statistical analyses of these angles.

## Statistical analyses

*t* Tests were used to assess the behavioral performance difference between the 2 types of trials (paired, 2-tailed) (Experiment 1 only), object decoding and cross-decoding performance against chance (1-sample, 1-tailed, as only effects above chance were meaningful), decoding performance difference between the 2 types of trials (paired, 2-tailed) (Experiment 1 only), cross-decoding drop (paired, 1-tailed, as the effect was expected to be either null or with cross-decoding being lower than within-decoding), and pairwise comparisons among the 3 ROI sectors (paired, 1-tailed as the existence of a more distractor-tolerant VWM representation in PPC than in OTC was explicitly tested based on prior research showing greater distractor resilience in VWM decoding in PPC than in EVC; [5–7]). Correction for multiple comparisons was

applied using the Benjamini–Hochberg method [83] for the number of tests of the same type performed within each ROI or sector, and for the 3 pairs of tests performed across the 3 ROI sectors. Repeated measures ANOVAs were used to assess the main effects and interactions across the 3 ROI sectors and pairs of ROI sectors.

## Supporting information

**S1 Fig. Behavioral performance.** (**A**) Experiment 1 behavioral VWM change detection performance. (**B**) Experiment 2 behavioral VWM change detection performance. Error bars indicate SE. Data are available from S1 Data and at osf.io/8rbkh/.
(PDF)

**S2 Fig. Experiment 1 vertex numbers and response time courses of each ROI.** (**A**) Average number of vertices in each ROI. Error bars indicate SE. (**B**) Time courses of beta weights for trials with and without distractors averaged over all the vertices within each ROI. In each ROI plot, the light gray vertical bars mark the stimulus presentation time during the encoding and probe periods, the medium gray vertical bars mark the fMRI decoding period during VWM delay, and the dark gray horizontal bar marks the distractor presentation time. See Materials and methods for more details. The lighter-colored ribbons around the plot lines represent SE. Data are available from S1 Data and at osf.io/8rbkh/.
(PDF)

**S3 Fig. Experiment 1 target and distractor decoding for trials with and without distractors.** (**A** and **B**) Target object decoding accuracy during VWM delay for trials with and without distractors for all the ROIs and for the 3 ROI sectors, respectively. The colored symbols above the bars mark the decoding significance of each bar compared to chance (.5). The black symbols mark the significance in decoding difference between trials with and without distractors. Error bars indicate SE. * $p < .05$, ** $.001 < p < .01$, *** $p < .001$. (**C**) Target and distractor decoding across time. In each ROI plot, the light gray vertical bars mark the stimulus presentation time during the encoding and probe periods, the medium gray vertical bars mark the fMRI decoding period for VWM delay, and the dark gray horizontal bar marks the distractor presentation time. See Materials and methods for more details. The horizontal dashed line indicates chance level decoding. The lighter-colored ribbons around the plot lines represent SE. Data are available from S1 Data and at osf.io/8rbkh/.
(PDF)

**S4 Fig. Simulation results from 10 cases.** Blue lines depict untransformed decoding accuracy as a function of the underlying signal strength. Orange lines depict probit-transformed decoding accuracy as a function of the underlying signal strength. See S1 Supplementary Results for more details.
(PDF)

**S5 Fig. Experiment 1 decoding time courses.** (**A**) Within- and cross-decoding over time of targets across trials with different types of distractors. (**B**) Within- and cross-decoding over time of targets across trials with and without distractors. In each ROI plot, the light gray vertical bars mark the stimulus presentation time during the encoding and probe periods, the medium gray vertical bars mark the fMRI decoding period for VWM delay, and the dark gray horizontal bar marks the distractor presentation time. See Materials and methods for more details. The horizontal dashed line indicates chance level decoding. The lighter-colored ribbons around the plot lines represent SE. Data are available at osf.io/8rbkh/.
(PDF)

**S6 Fig. Experiment 1 angles of target and distractor representations across trials with different types of distractors for each ROI.** (**A**). Representational space for targets and distractors with different target-distractor pairing. Each representational space geometry is an MDS projection of the group-averaged RDM of the 4 conditions included. Here, 2 types of targets (pink and purple) are paired with 2 types of distractors (black and grey lines). In a given trial, only 1 target (pink or purple) is shown with 1 type of distractors (black or grey line) (see Fig 1A). See main text for more details. (**B**) Target-target angles and (**C**) distractor-distractor angles. In each plot, angles calculated from the RDMs of the individual participants are shown on the left and those from the group RDMs are shown on the right. See main text for more details. Error bars indicate SE. Data are available from S1 Data and at osf.io/8rbkh/. (PDF)

**S7 Fig. Experiment 1 angles of target and distractor representations across trials with and without distractors for each ROI.** (**A**) Representational space for targets and distraction conditions, as in S6A Fig. (**B**) Target-target angles and (**C**) distraction condition angles, as in S6B and S6C Fig. In each plot, angles calculated from the RDMs of the individual participants are shown on the left and those from the group RDMs are shown on the right. Error bars indicate SE. Data are available from S1 Data and at osf.io/8rbkh/. (PDF)

**S8 Fig. Experiment 2 vertex numbers and response time courses of each ROI.** (**A**) Average number of vertices in each ROI. Error bars indicate SE. (**B**) Time courses of beta weights over all the vertices within each ROI. In each ROI plot, the light gray vertical bars mark the stimulus presentation time during the encoding and probe periods, and the medium gray vertical bars mark the fMRI decoding period for VWM encoding and delay periods. See Materials and methods for more details. The lighter-colored ribbons around the plot lines represent SE. Data are available from S1 Data and at osf.io/8rbkh/. (PDF)

**S9 Fig. Experiment 2 presentation order decoding.** (**A**) An illustration of the presentation order decoding. Here, a decoder is trained to decode trials containing the same 2 target objects but in different presentation orders. (**B** and **C**) Order decoding accuracy during VWM encoding and delay for all the ROIs and the 3 ROI sectors, respectively. The colored symbols above the bars mark the decoding significance of each bar compared to chance (.5). The black symbols mark the significance in decoding difference between the encoding and delay periods. The horizontal dashed line indicates chance level decoding. Error bars indicate SE. * $p < .05$, ** $.001 < p < .01$, *** $p < .001$. (**D**) Time courses of order decoding for all the ROIs. In each ROI plot, the light gray vertical bars mark the stimulus presentation time during the encoding and probe periods, and the medium gray vertical bars mark the fMRI decoding period for the VWM encoding and delay periods. See Materials and methods for more details. The horizontal dashed line indicates chance level decoding. The lighter-colored ribbons around the plot lines represent SE. Data are available from S1 Data and at osf.io/8rbkh/. (PDF)

**S10 Fig. Experiment 2 target within- and cross-decoding across time.** In each ROI plot, the light gray vertical bars mark the stimulus presentation time during the encoding and probe periods, and the medium gray vertical bars mark the fMRI decoding period for the VWM encoding and delay periods. See Materials and methods for more details. The horizontal dashed line indicates chance level decoding. The lighter-colored ribbons around the plot lines represent SE. Data are available at osf.io/8rbkh/. Note for this figure: The single probe object shown at the end of the delay was randomly chosen between the 2 target object types for both

change and no change trials (i.e., it was either an exact repeat or another exemplar from the same type of objects). Across trials in the same condition, efforts were not made to show an equal number of each of the 2 target object types as probes. This may explain why during the probe period (the second peak in S10 Fig) cross-decoding may be lower than within-decoding for the target objects; i.e., to decode AC vs. BC, if C does not appear equally often as probes in AC and BC trials, C may contribute to AC vs. BC decoding at the probe stage, resulting in a cross-decoding drop when the AC-BC trained decoder is used to decode AD vs. BD at the probe stage.
(PDF)

**S11 Fig. Experiment 2 angles of target-target representations during VWM delay for each ROI.** (**A**) Representational space for targets. Each representational space geometry is an MDS projection of the group-averaged RDM of the 4 conditions included. Here, 2 types of targets (pink and purple) are paired with 2 other types of target (black and gray lines). In a given trial, 1 target (pink or purple) is shown with another target (black or gray line) (see Fig 2A). See main text for more details. (**B**) Target-target angles. In each plot, angles calculated from the RDMs of the individual participants are shown on the left and those from the group RDMs are shown on the right. See main text for more details. Error bars indicate SE. Data are available from S1 Data and at osf.io/8rbkh/.
(PDF)

**S12 Fig. Experiment 2 angles of target-target representations during VWM encoding for each ROI.** (**A**) Representational space for targets, as in S11A Fig. (**B**) Target-target angles, as in S11B Fig. In each plot, angles calculated from the RDMs of the individual participants are shown on the left and those from the group RDMs are shown on the right. See main text for more details. Error bars indicate SE. Data are available from S1 Data and at osf.io/8rbkh/.
(PDF)

**S1 Supplementary Results. Simulation of decoding performance and signal strength.**
(DOCX)

**S1 Data. This file contains the measurements from the individual participants from which the means and errors were derived and depicted in all the data plots.**
(XLSX)

## Acknowledgments

I thank SuKeun Jeong for generating the visual stimuli, Judy Young Hye Kwon, Hillary Nguyen, and Ben Swinchoski for assistance in fMRI data collection, and Marvin Chun for general support. I also thank Roxana Ismail-Beigi for proofreading an earlier draft of the manuscript.

The content is solely the responsibility of the authors and does not necessarily represent the official views of the National Institutes of Health.

## Author Contributions

**Conceptualization:** Yaoda Xu.

**Data curation:** Yaoda Xu.

**Formal analysis:** Yaoda Xu.

**Funding acquisition:** Yaoda Xu.

**Investigation:** Yaoda Xu.

**Methodology:** Yaoda Xu.

**Project administration:** Yaoda Xu.

**Resources:** Yaoda Xu.

**Software:** Yaoda Xu.

**Supervision:** Yaoda Xu.

**Validation:** Yaoda Xu.

**Visualization:** Yaoda Xu.

**Writing – original draft:** Yaoda Xu.

**Writing – review & editing:** Yaoda Xu.

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
