## [Decision Letter · Decision Letter 0]

28 Aug 2024

Dear Dr Xu, 

Thank you for submitting your manuscript entitled "Untangling representations in visual working memory in human occipitotemporal and posterior parietal cortices" for consideration as a Short Reports by PLOS Biology.

Your manuscript has now been evaluated by the PLOS Biology editorial staff as well as by an academic editor with relevant expertise and two independent reviewers. 

As you know from the previous submission, we have a two-step submission system. Unfortunately, I forgot to ask you to fully submit your manuscript and provide the metadata after I discussed your revision with the Academic Editor. We have sent your manuscript out to review and have already received the reviewer reports (pointing to a Major Revision), but for procedural reasons, I still need to ask you to complete your submission by providing the metadata before I can send you the decision letter. To this end, please login to Editorial Manager where you will find the paper in the 'Submissions Needing Revisions' folder on your homepage. Please click 'Revise Submission' from the Action Links and complete all additional questions in the submission questionnaire.

To provide the metadata for your submission, please Login to Editorial Manager (https://www.editorialmanager.com/pbiology) within two working days, i.e. by Aug 30 2024 11:59PM.

Apologies for this slightly complicated process, but once you've submitted the full manuscript, I will immediately be able to send you the formal Major Revision letter.

Kind regards,

Christian

Christian Schnell, PhD

Senior Editor

PLOS Biology

cschnell@plos.org

---

## [Editor Report · Decision Letter 1]

5 Sep 2024

Dear Yaoda,

Thank you for your patience while we considered your revised manuscript "Untangling representations in visual working memory in human occipitotemporal and posterior parietal cortices" for publication as a Short Reports at PLOS Biology. Your revised study has been evaluated by the PLOS Biology editors, the Academic Editor, one of the original reviewers and one new reviewer (Reviewer 2)

In light of the reviews, which you will find at the end of this email, we would like to invite you to revise the work to thoroughly address the reviewers' reports.

As you will see below, the reviewers agree that your study has significantly improved through the revision. However, there are still a few concerns that need addressing, including the interesting and important point about filtering distractors at or prior to PPC raised by Reviewer 2, ideally with experimental data.

Given the extent of revision needed, we cannot make a decision about publication until we have seen the revised manuscript and your response to the reviewers' comments. Your revised manuscript is likely to be sent for further evaluation by all or a subset of the reviewers.

**IMPORTANT - SUBMITTING YOUR REVISION**

*Re-submission Checklist*

*Published Peer Review*

*PLOS Data Policy*

*Blot and Gel Data Policy*

Sincerely,

Christian

Christian Schnell, PhD

Senior Editor

PLOS Biology

cschnell@plos.org

REVIEWS:

Reviewer 1:

"Untangling representations in visual working memory in human occipitotemporal and posterior parietal cortices," Xu. This is a revision, and overall I find it to have been very responsive (and the point-by-point cover letter to be very clear and thorough.) In particular, removing the original Figure 1 removed some confusion, and adding the MDS visualizations and the calculation of angles separating representational subspaces now bolsters the interpretation of the decoding results, which support only indirect inference about representational geometry.

The author writes:

"In my review of the literature (including the references that the reviewer has provided),

orthogonal target-distractor and target-target representations in VWM have not been

previously established in the human PPC. Among the references that the reviewer

kindly provided, a majority of them addressed issues that are not the focus of the

present investigation, such as ... As such, it is fair to say that previous research has not rigorously investigated and firmly established the orthogonality of target-distractor and target-target representations in VWM in the human PPC."

I don't disagree with this concluding statement, but would also suggest that this is a rather narrow (and very specific) focus. If the author's goal with this paper is to communicate her findings with other scientists studying related questions, an effective way to broaden the potential impact of this work might be to make reference to other work in which a similar scheme (orthogonal representation) has been proposed for somewhat different contexts. E.g., how targets in VWM with different priorities may be coded during VWM delay; attention-related, but not VWM-related, processing ; the rotation between sensory input and LTM-stored information (Libby & Buschman, 2021); and interaction between attention and VWM (Panichello & Buschman, 2021) are all examples of visual cognition, and although what this author is reporting here is strictly limited to "orthogonality of target-distractor and target-target representations in VWM in the human PPC," what she reports will also be of interest to people studying these other kinds of visual cognition.

Below I'll add some minor points that the author may want to consider that could help with clarity.

p. 4. "entanglement" Please consider being more explicit about the context in which it is being used, e.g., actually include something along the lines of "I borrowed the term from the object processing literature, in which untangled representations would allow independent read-out of the different representations, whereas tangled representations would not." And then also include citations that support this.

Fig. 1.J. These values are not from a time series, but are discrete values, one from each ROI, and so should be plotted discretely (e.g., with same format as the other bar graphs in this figure).

Legend to Figure 1. Here, and elsewhere, is it more precise to use "… trained to classify bikes relative to hangers" or "… trained to discriminate bikes from hangers"?

Legend to Figure 1. Panels J and K are not included in the legend.

p. 8. Consider specifying that "Significant VWM decoding OF THE TARGET was obtained …", so as to clarify that this is item decoding, rather than decoding of pairings of stimuli.

p. 13: "Thus, only in IPS2-IPS4 we have near 0 … Consider rephrasing to be less colloquial?

p. 13: MDS results don't confirm the cross-decoding results so much as they support your interpretation of those results.

Figure 4 and related text: it would be more straightforward to present the encoding results first, then the delay results.

"Participants" section: the "Yu … X (20xx)" citations should be formatted with "et al." like the others?

Supplementary Figure 3: I found legend difficult to follow, particularly because the 'encoding' bar is difficult to see. Consider including an empty plot that uses written labels and arrows to show what shading and "distraction timeline" correspond to (i.e., almost like a legend for the graphical conventions)?

Reviewer 2:

This is an interesting topic, the data are high quality, and the experiment was well-executed with multiple sessions collected from n=14. That said, I found the logic of the analyses and the interpretation of the data to be a bit challenging to follow.

I think the author’s interpretation of Experiment 1 boils down to this - (1) in OTC representations of specific target categories change as a function of the distractor category they are paired with, therefore target-distractor representations are entangled, and (2) in IPS2-4 target representations do not change as a function of the distractor they are paired with, therefore target-distractor representations are disentangled.

If this is the intended interpretation, I would find the paper much easier to digest if simpler terminology were adopted. For instance, nowhere do you need to say “orthogonal”, and you could probably get rid of “entangled” and “disentangled” and just say “interacting” and “independent” or something of that sort. This seems like a minor suggestion, but I think it is quite important, especially for a general readership journal and people outside the field are going to struggle. I am quite familiar with this general field, and the work of DiCarlo and Rust and others, and I found that there wasn’t a clear statement about the rationale/predictions that link all these terms together in a meaningful way. So, my suggestion is to simplify all the jargony descriptions and to have a very clear statement that spells out the logic of the analysis step-by-step before diving into the results.

The “orthogonal” target-distractor representations in PPC are described as if the orthogonalization occurs via some active and adaptive process. I think an alternative explanation is that PPC just doesn’t encode much information about the distractors and therefore the representations don’t interact. Supplemental Figure 3C seems to support this, as in higher areas of IPS distractor decoding is about on par with target decoding. In contrast, distractor decoding in all earlier areas is higher than target decoding. What supports the “adaptive orthogonalization” account over an account that posits filtering of distractor information (the alternative account you set up in the intro)? I understand that simulations were performed to address the SNR issue, but here I’m focused not on overall SNR but on the relative SNR of target and distractor representations. Distractors dominate in OTC but are on par with targets in PPC. Thus, it seems like supporting the “adaptive orthogonalization” account over the “distractors are filtered out before PPC” account would require some evidence that the target-distractor representations start entangled and then actively orthogonalize. Alternatively, you could show that the independence holds across different levels of distractor decoding accuracy. Otherwise, I’m not sure how to tease apart these two accounts?

The presence of distractors has no impact on target decoding accuracy. So whatever interactions or entanglement exists between targets and distractor categories doesn’t degrade the amount of information about the targets. This seems consistent with a simpler interpretation that early visual areas care more about low-level features and spatial configurations of stimuli, whereas PPC is less selective for these attributes? Again, maybe this is the point the author is trying to make – that distractors have been filtered out before PPC and so PPC is more invariant or stable. However, I am again a bit confused about why the conclusion isn’t just that early visual areas are more sensitive to low-level sensory features whereas PPC is more sensitive to relevant features that are attended or stored in memory (consistent with many prior studies). However, in both OTC and PPC, distractors don’t lead to a loss of information about the remembered target.

Three more comments just to consider.

1) I think you added this text in response to a prior reviewer comment, but I would not suggest including your alternative account of another’s work based on speculation about the methods that they used in a non-peer-reviewed version of their paper (Degutis). This seems unnecessary as their paper is not published could prove awkward down the road if your presumption is incorrect. You could email them to ask, or to see if they have an updated version – especially a version of record – but I would personally endorse removing that passage.

2) This may reflect something that the other reviewers know and I am missing, but I don’t think the angle measurements add much beyond the cross-decoding analysis. If there is a drop in cross-decoding, there is a rotation, if not, then no rotation. Perhaps removing some of the “orthogonal” jargon would make these analyses seem even less necessary. Again, apologies if I’m missing something.

3) Power is more than the number of subjects – it is a function of the number of subjects and trials within each subject. Just mentioning because this is a well-powered study but your justification only focuses on sample size.

---

## [Editor Report · Decision Letter 2]

22 Oct 2024

Dear Dr Xu,

Thank you for your patience while we considered your revised manuscript "Untangling representations in visual working memory in human occipitotemporal and posterior parietal cortices" for publication as a Short Reports at PLOS Biology. This revised version of your manuscript has been evaluated by the PLOS Biology editors and the Academic Editor.

Based on our Academic Editor's assessment of your revision, we are likely to accept this manuscript for publication, provided you satisfactorily address the following data and other policy-related requests:

* We would like to suggest a different title to improve the accessibility of your title: "The human occipitotemporal and posterior parietal cortices orthogonalize the representation of different streams of information concurrently coded in visual working memory"

* Please add the links to the funding agencies in the Financial Disclosure statement in the manuscript details.

* In the ethics statement, please (i) provide information on whether the participants provided written or oral consent, (ii) provide the approval number of the ethics committee and (iii) provide information that the study has been conducted in accordance with the principles of the Declaration of Helsinki.

* DATA POLICY:

Regardless of the method selected, please ensure that you provide the individual numerical values that underlie the summary data displayed in the following figure panels as they are essential for readers to assess your analysis and to reproduce it: 1EFHIJK, 2CDFG, 3CDEF, 4BD, S1AB, S2A, S3AB, S6BC, S7BC, S8A, S9BC, S11B and S12B.

* CODE POLICY

We expect to receive your revised manuscript within two weeks. 

*Published Peer Review History*

*Press*

Sincerely,

Christian

Christian Schnell, PhD

Senior Editor

cschnell@plos.org

PLOS Biology

---

## [Editor Report · Decision Letter 3]

25 Oct 2024

Dear Yaoda,

Thank you for the submission of your revised Short Reports "The human occipitotemporal and posterior parietal cortices orthogonalize the representation of different streams of information concurrently coded in visual working memory" for publication in PLOS Biology. On behalf of my colleagues and the Academic Editor, Ed Vogel, I am pleased to say that we can in principle accept your manuscript for publication, provided you address any remaining formatting and reporting issues. These will be detailed in an email you should receive within 2-3 business days from our colleagues in the journal operations team; no action is required from you until then. Please note that we will not be able to formally accept your manuscript and schedule it for publication until you have completed any requested changes.

PRESS

Sincerely, 

Christian

Christian Schnell, PhD

Senior Editor

PLOS Biology

cschnell@plos.org